# OpenReview forum: "Do Influence Functions Work on Large Language Models?"
_ICLR.cc/2025/Conference — ICLR 2025 Conference Withdrawn Submission_

### Official Review · Reviewer_Rsbz · 2024-10-17

**Soundness:** 2
**Presentation:** 3
**Contribution:** 2
**Rating:** 5
**Confidence:** 3

**Summary:**

This paper investigates the application of influence functions on LLMs. Influence functions aim to measure the impact of training data on the loss of specific validation samples, potentially enabling the study of decision-making in LLMs (e.g., bias detection) or analyzing training data for high-quality or adversarial samples. However, the paper finds that influence functions currently do not work for LLMs due to three main reasons: (1) inaccurate approximation of the Hessian matrix, (2) unconverged LLMs, and (3) an inappropriate definition of influence functions based on gradients.

**Strengths:**

- The paper is well-written and easy to follow.
- The plots and tables provide relevant and interesting data that support the conclusions.
- The analysis is thorough, with several interesting results.
- The code is included in the supplementary material, and after reviewing it, I am confident that the results can be reproduced, and the implementation is accurate.

**Weaknesses:**

- The paper presents only negative results without offering alternative approaches. For example, it would be beneficial to discuss alternative definitions that do not rely on gradients. While RepSim does not fit the traditional definition of gradient-based influence functions, it does show promising results and could be considered as an alternative. Thus, on one hand, the authors are calling for alternative methods to quantify influence, and on the other they are showing that RepSim, which is such an alternative, performs great on all their benchmarks. This seems a contradiction.
- The experiments are conducted on small datasets which are much smaller than typical LLM training data. It is unclear whether RepSim would still outperform other methods on a larger scale. This could also be a fourth reason why influence functions struggle with LLMs, as identifying the most influential samples might be more challenging with larger datasets. Unfortunately, this aspect is not discussed.
- DataInf, LiSSA, and GradSim are not the only methods for estimating influence functions. A quick search reveals two other methods ([1] and [2]) that seem to perform well. In fact, [1] suggests they outperform RepSim. However, no comparison is made with these methods, making it unclear if the conclusions generalize beyond the three tested methods.
- The experiments use only a single, outdated model, Llama-2, making it unclear whether the findings are specific to this model or if they generalize to other LLMs.

Overall, I am concerned that the conclusions may not generalize beyond the three methods tested, the single model used, and the small datasets employed. Additionally, I question whether RepSim should be considered something that "should be outperformed," as it could be seen as a different type of influence estimator. Of course, similarity is not the same as influence, but given that the evaluated applications of influence estimation all seem to be covered by RepSim, there does not seem to be a reason to make this distinction based on the current experiments.

My current score is based on an expectation that the authors are able to address some of my concerns to the expected degree. However, if they cannot, I will reduce my score. In contrast, if they can additionally clarify some aspects, especially with regards to RepSim and the missing baselines, I would also increase my score.

[1] Choe, Sang Keun, et al. "What is Your Data Worth to GPT? LLM-Scale Data Valuation with Influence Functions." arXiv preprint arXiv:2405.13954 (2024).

[2] Grosse, Roger, et al. "Studying large language model generalization with influence functions." arXiv preprint arXiv:2308.03296 (2023).

**Questions:**

- In the code, the models are loaded in 8-bit format. I am concerned that this could affect gradient computation, potentially leading to worse gradient estimates for the methods presented. Did the authors check whether this affected performance? Could they include an ablation study to ensure that 8-bit instead of 32-bit gradients are not the cause of the observed performance degradation?
- Could the authors provide an example where RepSim would not be able to solve an interesting application of influence estimation? If they can provide one, could they also evaluate the influence estimators on this application instead? How would one show that current influence estimation methods do not work in this case?

---

### Official Review · Reviewer_LyU3 · 2024-10-31

**Soundness:** 2
**Presentation:** 3
**Contribution:** 2
**Rating:** 3
**Confidence:** 4

**Summary:**

The paper Do Influence Functions Work on Large Language Models? investigates the efficacy of influence functions in the context of large language models (LLMs). Influence functions, designed to assess the impact of individual training data points on model outputs, have shown promise in simpler models but are largely ineffective when extended to LLMs.

The authors conduct a series of experiments applying influence functions to three primary tasks: harmful data identification, class attribution, and backdoor poison detection. Their results demonstrate that influence functions fail to reliably identify harmful training data, isolate triggers for backdoor attacks, or accurately determine data points contributing to model performance.

The paper further discusses potential reasons for these shortcomings, including the approximation errors associated with the inverse Hessian-vector product (iHVP) matrix, the fact that fine-tuned LLMs may not settle into global minima, and the challenges of associating parameter changes, with meaningful alterations in model behavior.

**Strengths:**

The authors present a comprehensive set of experiments across diverse tasks to clearly demonstrate the ineffectiveness of influence functions when applied to LLMs. In addition to the empirical evidence, the paper offers hypotheses to explain the underlying causes of this ineffectiveness.

Overall, the manuscript is well-structured and easy to follow, with the exploration of influence functions in the context of LLMs being both timely and compelling.

**Weaknesses:**

Major Concerns:

1. Lack of Critical Details: The paper is missing essential details, particularly in the experimental setup described in Section 3. The authors only mention the dataset and model used but do not explain how they compute the value of the influence function. This omission raises several questions:

  - Gradient Calculation: How did the authors compute the gradients? Specifically, did they treat all parameters across different layers as a single $\theta$, or did they calculate gradients selectively for specific layers? If they used the former approach, how did they manage the complexities of LoRA and fit the massive gradient matrix into a 40GB A100 for computation?
  - Loss Function Choice: How did the authors choose an appropriate loss function given the sequence generation nature of large language models (LLMs)? Additionally, how did they define the (x,y) pairs for use in the influence function?
  I also attempted to access the GitHub repository provided by the authors, but it appears to be unavailable.

2. Unreasonable Experimental Logic: The experimental design lacks coherence, especially in the harmfulness experiment. The influence function is supposed to measure the impact of individual training samples on the loss function, which the loss function in this context, should represent some measure of harmfulness. This ties back to the first concern: without a clear definition of the loss function, the entire basis of the harmfulness experiment is unclear. The same issue arises in the other two experiments.

3. Unconvincing Explanations for Failures: The explanations offered for potential reasons behind the failures are not compelling.

  - Uncertain Convergence State: The authors could refer to [1], where similar convergence violations are discussed but with different conclusions.
  - Effect of Parameter Changes: The authors only demonstrate that the L2 norm of $\Delta \theta$ does not correlate with the attack success rate (ASR). However, this merely shows that the L2 norm is ineffective and does not provide evidence against the validity of the influence function. The utility of the influence function lies in its problem-specific nature and the associated loss landscape, which differentiates it from simple metrics like the L2 norm of $\Delta \theta$.

Minor Concern:
- The definition of "Acc" on the y-axis of Figure 8 needs to be clarified.

Reference:
[1] Schioppa, Andrea, et al. "Theoretical and practical perspectives on what influence functions do." Advances in Neural Information Processing Systems 36 (2024).

**Questions:**

The authors need to respond to the points in Weaknesses section.

---

### Official Review · Reviewer_FC7d · 2024-11-03

**Soundness:** 2
**Presentation:** 2
**Contribution:** 3
**Rating:** 3
**Confidence:** 3

**Summary:**

This paper investigates whether influence functions work effectively on LLMs. Through empirical evaluation across multiple tasks, the authors find that influence functions consistently perform poorly in most settings. Their investigation reveals three main reasons for this underperformance: (1) inevitable approximation errors in estimating the inverse Hessian-vector product (iHVP) due to LLM scale, (2) uncertainty in convergence during fine-tuning, and (3) a fundamental limitation in the definition itself, where changes in model parameters don't necessarily correlate with changes in LLM behavior. These findings indicate the need for alternative approaches to identify influential samples.

**Strengths:**

- Provides comprehensive empirical investigation of influence functions in LLMs
- Systematically analyzes and identifies key failure modes
- Offers both practical and theoretical insights into limitations

**Weaknesses:**

- Limited Motivation:
- - The complexity and computational aspects of iHVP approximation are not fully discussed
- - Benefits compared to existing data filtering methods (such as those in Dolma paper: https://arxiv.org/pdf/2402.00159) are not well justified
- Methodological Clarity:
- - AdvBench task specifications could benefit from example illustrations
- -  RepSim's methodology and its strong performance lack detailed explanation
- - The rationale for using different datasets for performance evaluation and causality analysis is unclear (e.g., MNIST)
- - No evaluation on downstream tasks such as in https://arxiv.org/pdf/2104.12567
- Analysis Scope:
- - Study is limited to Llama as the backbone model
- - Section 4.2's findings raise questions about influence function behavior in smaller language models
- - Relationship between uncertainty and factors like data, model size, or fine-tuning could be explored further
- - Potential connections to pretraining effects not discussed

**Questions:**

See weaknesses

---

### Official Review · Reviewer_VBHP · 2024-11-04

**Soundness:** 2
**Presentation:** 2
**Contribution:** 2
**Rating:** 5
**Confidence:** 2

**Summary:**

This paper investigates whether (approximated) influence function methods can be applied to large language models.
In the first part of the paper, three influence function methods (LiSSA, DataInf, GradSim) and one gradient-similarity method (RepSim) are applied to three applications (1. Harmful data identification; 2. Class attribution; 3. Backdoor poison detection). It is shown that the three influence function methods perform poorly in these three applications.
In the second part of the paper, various experiments were conducted to show that the poor performance can be attributed to (1) approximation errors; (2) uncertain convergence states; (3) the small correlation between parameter change and model behavior change.
To conclude, the paper advocates for alternative approaches for influential data identification, and a re-evaluation of influence function methods.

**Strengths:**

* The topics studied in this work are important and interesting. Identifying influential examples in model training is an important problem, and investigating and questioning influence function based methods from prior works may be of interest to many researchers.
* Comprehensive experiments covering various methods, applications, and to support the analysis of potential reasons behind the poor performance.
* In particular, I agree with the third reason attributing to the poor performance. Influence function methods are based on loss, however oftentimes we care about metrics other than loss (e.g., attack success rate) that are not directly reflected by the loss, making influence function methods less useful.

**Weaknesses:**

* I have concerns with the paper presentation. I find this paper confusing at various points.
  * Background information and experiment settings are not explained clearly.
  * Tables and figures are not self-contained or explained clearly. The connection between the tables/figures and the main text is weak and may be strengthened.
  * See my questions below.

(Please be aware that influence function is not my direct research area and I am only familiar with the basic concepts; I gave a confidence score of 2)

**Questions:**

General questions
* What are the core concepts or innovations of LiSSA, DataInf, GradSim. Is it possible to put them into a unified view and explain their differences?
  * Additionally, why does the computation overhead of LiSSA grow with more epochs?
* In line 134 it was mentioned that “they often suffer from limited evaluation settings and lack of robust baselines for comparison.” How are influence function works usually evaluated? What are the limitations in those works exactly?
  * Additionally, is the evaluation used in this paper (acc and coverage) used in these past works or commonly adopted in general?
* I’m concerned with the “class attribution” experiment. Is it possible that an influential example comes from a _different_ class and it is important because it pushes the decision boundary to the right place? Correct me if I’m misunderstanding something here.

Specific questions
* How is “attack success rate” defined in Section 3.1.
* In line 188, what does it mean to use a BERT-style classifier to evaluate the “attack success rate”? How does it work?
* In Figure 2,
  * The subfigure of LiSSA and GradSim looks exactly the same? Is there any reason behind this?
  * It seems that there is a dominant example. Is it the one shown in Figure 1? any explanation why it is given a high score by all influence function methods?
* Is it possible to conduct leave-one-out validation with the small mixed data setting (40 data points)? This will provide good reference to section 3.1.

Presentation suggestions:
* In Table 2 or its caption, describe what “small/large data mix” means
* In Figure 2, highlight the harmful examples with a line or a box.
* In Figure 3, describe what 3x3 and 20x20 means.
* Line 459, “to analysis” -> “to analyze”
* Line 510, “thier” -> “their”

---

### Note · Authors · 2024-11-29

I have read and agree with the venue's withdrawal policy on behalf of myself and my co-authors.